# The Stakeholders’ Involvement in Healthcare 4.0 Services Provision: The Perspective of Co-Creation

**DOI:** 10.3390/ijerph20032416

**Published:** 2023-01-29

**Authors:** Norbert Laurisz, Marek Ćwiklicki, Michał Żabiński, Rossella Canestrino, Pierpaolo Magliocca

**Affiliations:** 1Department of Public Management, Cracow University of Economics, 31-510 Cracow, Poland; 2Department of Management and Quantitative Studies, Parthenope University of Naples, 80133 Naples, Italy; 3Department of Humanities, Faculty of the Humanities, Literature, Cultural Heritage, and Educational Sciences, University of Foggia, 71122 Foggia, Italy

**Keywords:** Health 4.0, cocreation, health system management, efficiency of health services, creation of healthcare products and services, involvement of patients and health workforce

## Abstract

Literature research on cocreation in healthcare indicates the theoretical sophistication of research on collaboration between healthcare professionals and patients. Our research continues in the new area of Health 4.0. Cocreation has become an essential concept in the value creation process; by involving consumers in the creation process, better results are achieved regarding product quality and alignment with customer expectations and needs. In addition, consumer involvement in the creation process improves its efficiency. Cocreation allows for more efficient diagnosis and treatment of patients, as well as better and more effective use of the skills and experience of the health workforce. Our main objective is to determine the scope and depth of the cocreation of health services based on modern technological solutions (Health 4.0). We selected four cases involving Health 4.0 solutions, verified the scale and scope of cocreation using them as examples, and used the cocreation matrix. We used literature, case studies, and interviews in our research. Our analysis shows that patients can emerge as cocreators in the value creation process in Health 4.0. This can happen when they are genuinely involved in the process and when they feel responsible for the results. The article contributes to the existing theory of service cocreation by pointing out the limited scope of patient involvement in the service management process. For cocreation in Health 4.0 to increase the effectiveness of medical services, it is necessary to implement the full scope of cocreation and meaningfully empower the patient and medical workers in the creation process. This article verifies the theoretical analysis presented in our team’s previous article.

## 1. Introduction

New technologies today provide the most significant boost to the development of medical products and services. As a result, the whole sector, the management of medical care, diagnosis, treatment, service and patient care or prevention is changing [1]. Healthcare institutions are introducing more advanced technologies to help better manage public health challenges [2]. The use of Internet and telecommunications tools and services and remote diagnostics is growing, while medics are increasingly using artificial intelligence (AI) support in prevention, treatment and diagnosis [3]. They impact not only medical procedures but also the way of service provision [4]. A new research area is developing, named Health 4.0 [5,6], after adopting technologies for Industry 4.0 or the fourth industrial revolution [7].

The increasing technical sophistication of the devices or treatment or diagnostic modalities is prompting a change in how these modern solutions are designed. In addition to the design, the changes should also concern the approach to marketing the devices or services in question. In this context, lowering the barrier to acceptance of Health 4.0 solutions is vital. And this effect can be achieved by using the experience and knowledge of those who use such solutions, i.e., patients and doctors. Users’ experience allows changes to be introduced even at the stage of design or production, thanks to which the product becomes more user-friendly and, at the same time, its quality, usability and, consequently, acceptance of new solutions increases. The introduction and involvement of users (consumers) in creating and implementing products are called cocreation [2]. The introduction of cocreation into the process of creating medical products and services, as proposed by R. Palumbo [1], should contribute to increasing the effectiveness of the products and services thus produced. It enables dealing with the growing problem of a shortage of medical professionals and medical services in the context of successively increasing demand for them [8].

Our work combines these two strands into one research design. The paper aims to identify the scope of engagement of different actors (stakeholders) of health services development driven by new technologies (Healthcare 4.0 services) from the cocreation perspective. Cocreation combines the areas of managing the value creation process, technologies for creating and implementing new solutions, improving the usability of products and services, and creating consistent relationships between consumers and the producer and product. Based on the systematic literature review’s results, we identified examples of cocreation in healthcare with new opportunities associated with the fourth industrial revolution. The theoretical framework verified in previous authors’ work “Co-creation in Health 4.0 as a new solution for a new era”—the cocreation matrix—revealed common areas due to cross-case comparison.

The cocreation perspective provides more in-depth insights about creating classic products and services aimed at a less precisely defined consumer. The study enriches the knowledge about cocreation. It results from the two-tier approach to the consumer in the case of medical products and services since the actual consumer is the patient. However, consumption takes place under the supervision and/or at the behest of doctors, who, like patients, are not part of creating products and services.

Our study is organised as follows. First, we define the area of analysis to be carried out: Health 4.0. Second, we briefly present and explain the issue of cocreation by focusing on the phases of the cocreation process and the actors involved in the process. We explain how the analysed cases were selected by referring to the literature review presented more broadly in our author’s theoretical article, “Co-creation in Health 4.0”, as a new solution for a new era. We next present the research method used. We further present each of the four cases using the cocreation matrix and assess the scope and scale of cocreation in each case. The result of our analysis is a practical test of the cocreation matrix; the effect is presented as a summary of the four cases at the end of the paper.

## 2. Literature Review

Our considerations are products and services in the area of Industry 4.0. These primarily include the Internet of Things (IoT) and artificial intelligence (AI) [7]. New technologies are changing the social, public and economic space [9]. In the public and economic context, the main direction of this change is the widespread digitization and automation of processes as well as the integration of coordination processes in complex systems, both intrasectoral (e.g., between market players) and intersectoral (e.g., public sector and market or social sector and market) [7,10,11]. The process of digitization and automation is accelerating, and its scale is constantly expanding [12].

The health and medical services sector is using new technologies to an increasing extent [13]. Medical procedures are making increasing use of new technologies: surgery uses robots for precision and remote operations [14]; modern implants and prostheses are available, e.g., bionic implants and prostheses printed with 3D printers [15,16,17]; analytical and diagnostic data are being massively exploited thanks to new technologies that collect and analyse such data, e.g., through the use of AI [18]; routine medical procedures use medical devices, e.g., real-time monitoring of the user’s health indicators to support and control the medical practice [19,20]. Industry 4.0, from the healthcare perspective, may be perceived as solutions for individuals (e.g., broadly defined eServices such as wearable technology; Internet of Medical Things) [21,22]; solutions for healthcare institutions (e.g., big-data for clinical trials) [23]; new medical treatment and health therapies (new medical procedures) [18]; intelligent computing (UAV computing) [24]; digital twins (DTs) [25]. We define Health 4.0 as healthcare operating under the new opportunities created by the fourth industrial revolution [7].

In the classical view, businesses treated consumers only as consumers, i.e., neutral participants in marketing a product, whereas nowadays, the consumer is becoming a key actor in the whole process. Science calls this consumer-centrism [24,25]. In developing a consumer-oriented approach, broad coproduction becomes the most critical element. However, this coproduction is not understood as so-called stakeholder coproduction but is a much broader concept and is considered on many levels. In the case of customer orientation, we speak of value coproduction, i.e., the involvement of stakeholders not only in creating a product or service but also in its functionality, usability, and assimilability, i.e., the broadly understood value [26]. From this perspective, the product creation process is only a component of the value creation process. If this process is extended to include stakeholders representing the demand side (users, consumers), it is called value cocreation [27]. A company, public entity or social organization creates a physical product or service, while the creation and cocreation of value take place through extensive collaboration with stakeholders; in the latter case, the key actor is the consumer, who becomes a cocreator who creates value [28,29,30,31,32,33]. In value creation and cocreation, the most important aspect is the active relationship/interaction between all actors: creator, producer, supplier and consumer. Cocreation becomes a way of creating new solutions in many market ventures, social activities or public policies [24,30,31,34,35,36].

Based on the research results, projects using value cocreation achieves economic success and success in the psychosocial dimension. This success consists of a growing brand and product loyalty and a high acceptance of innovation and new solutions [26,37]. An additional benefit is the involvement of consumers in the process, which translates into more than enough work for better results. This is particularly evident in the case of solutions in the area of health, where consumers work not only to improve the functionality of a product or service but also for their health [38]. The effectiveness and sustainability of value creation in the area of new medical techniques and solutions carried out in this way is dependent on easy and open access to the entire process [26,39], on the network of stakeholders involved in the process [40], and on the flow of information and active communication between them and the companies [38,41]. Such broad collaborations that draw on consumer knowledge and experience result in more readily accepted and assimilated products by doctors and patients [42]. In summary, broad and multifaceted collaboration with stakeholders—mainly doctors, medical staff and patients—is a prerequisite for an effective Health 4.0 product development process [43]. Research and business practice show that cocreation helps providers succeed in creating new Healthcare 4.0 services [36,39,44,45].

Creating new value (products or services) is a process that follows a specific procedure and a specific pattern [46]. The specificity of the process, the way of proceeding and the sequence of procedures implemented may differ in each case. However, the very involvement of stakeholders, especially stakeholders representing the demand side of later transactions (in the case of health, it refers to patients and doctors/medical staff), is necessary and essential [27,34,36,44]. In the case of modern medical products and services, the value creation process proceeds in a specific way, so below, we provide a brief overview of the phases and actors of this process, considering the specifics of Health 4.0.

We use a modified life cycle development [43] classification for cocreation analysis where the value creation process consists of four phases (see Table 1).

The healthcare and medical services system includes medical personnel, mainly doctors and nurses, and external stakeholders such as labour unions, insurance companies or public insurers, and companies that produce pharmaceuticals or medical products and services [40]. Based on the literature review, we classified all the actors as regulators, providers, payers, suppliers and patients as J. Bessant, C. Künne and K. Möslein have proposed [48] (Table 2).

Three research questions guided our analysis: (1) Is cocreation used to create Health 4.0 products and services? (2) What is the scope and scale of cocreation in the studied cases of Health 4.0 products and services? (3) Do the cases in which we analyse differ in terms of how cocreation is used to create a service/product?

## 3. Research Methodology

The research design composes of the systematic literature review protocol and cross-case comparison based on pattern matching technique. First, we identify the cases where cocreation is investigated in services driven by new technology. For this purpose, we have searched Scopus, Web of Science and PubMed databases using the keywords “co-creation,” “Health 4.0” or “eHealth”. We identified 57 nonrecurring literature records using the time criterion of publication between 2010 and 2021. We excluded records that did not correspond to cocreation and Health 4.0/eHealth simultaneously. We excluded 26 articles due to substantive or formal deficiencies during the record screening stage of our literature review. This step is used to verify articles for substantive or methodological matching to the issue under analysis. In the next step, we excluded eight articles due to content limitations and insufficient data to fill the cocreation matrix. Finally, 23 articles were qualified for content analysis. The selection criteria included analysing the different phases of cocreation and the degree of involvement of actors in the phases in question. We identified three cases that best related to the keywords we were looking for and, at the same time, met the expectations of the investigators. We have added one case of a Health 4.0 solution implemented in Poland, TeleAngel (Table 3). Additionally, we searched grey literature for new evidence not covered in scientific publications, such as documents, company data, project groups and interviews published in the national language.

The identified cases are presented in the same manner. First, we explain what technology was used and its specifics; next, what actors were involved and what areas of co-creation are concerning the critical phase of the product/service life cycle.

Finally, we conducted a cross-case comparison using the cocreation matrix. Three researchers coded all cases. Discrepancies were discussed, and the results were matched to a cocreation framework. To summarise each case, cocreation matrices were created (Table 4, Table 5, Table 6 and Table 7). These tables synthesise the study’s results, verifying the assumptions about the full use of cocreation during the product/service development process and providing an actual picture of its use in a specific case. Figure 1 completes the analysis; it contains the combined results of all the cases studied.

## 4. Results

### 4.1. Cases Presentation

During the literature review, we selected four cases to analyse the scale and scope of cocreation in developing a Health 4.0 product/service. Each case was described in the scientific literature, and the baseline characteristics of each case are briefly presented below (Table 3). Further description of each case complements the analysis presented in the scientific articles. Other research and analysis, as well as interviews and case descriptions available in the public space, were used for this purpose.
ijerph-20-02416-t003_Table 3Table 3Synthesis of cases.Case Number#1#2#3#4**Name**Emergency Medical Service SystemDigital Health Platform Paginemediche.itElectronic Medical RecordMalopolska TeleAngel**Country**ThailandItalyItalyPoland**Year**2015201920152018**Research design**QuantitativeQuantitativeQualitativeQualitative**Research method**SurveySurveyCase study; interviewsCase study; document analysis**Technology used**Mobile phones (mobile and cloud technology)Internet communication platformInternet/mobile communication platform; barcodeRemote health monitor using an electronic wristband**Area of implementation**Communication between doctors, nurses–patients and patients’ familiesCommunication doctors–patientsCommunication doctor–doctorHealth monitoring of older peopleto identify illness and therapy**Research subjects**40 medium-to-large-sized of hospitals293 doctors using the engagement platform servicesAcademic Integrated Hospital (AIH); 10 AIH employees Project stakeholders**Main source**Sukkird and Shirahada, 2015 [50]Lo Presti et al., 2019 [42]Bonomi et al., 2015 [51]Ćwiklicki and Żabiński, 2018 [52]Source: own elaboration.


#### 4.1.1. Case #1: Emergency Medical Service System (EMSS)

##### Description

The Emergency Medical Service System (EMSS) is defined as “a specific arrangement of emergency medical professionals, equipment and supplies designed to function in a coordinated manner” [50]. The system’s design relies on volunteers to support the online healthcare, emergency services, emergency medical service and medical transport system [53]. As a result of the inability to fully fund the emergency medical system, a hybrid solution was designed. The launch of online support has resulted in lower admissions to emergency departments. Direct medical assistance, on the other hand, works more efficiently and quickly. Emergency medical assistance is provided by medically trained volunteers who use their own or also voluntarily provided communication channel, medical equipment or means of transportation to care for the sick and diagnosis or transport to hospitals [54]. The Thai emergency medical and care system is an exciting example of combining top-down funding for direct medical activities in centres and using volunteers at the micro level—in direct contact with the patient, in the role of caregiver, helper or paramedic.

##### Actors

Sukkird and Shirahada [50] distinguished the following participants: consumers of services: injured persons, family members and caregivers; providers of services, which can be divided into healthcare providers (medical staff, hospital) and IT providers (mobile company, technical support centre). Payers are the insurance company and EMS agent. Direct providers are volunteer family members and caregivers.

##### Mechanisms/Areas of Cocreation

A necessary means to the overarching value: of “active ageing patients” is communication represented by technology transfer and knowledge sharing—the flow of information [55]. The actors on the right represent this aspect. On the left, we have service providers, mainly referred to as the flows of technology, knowledge or professional service [56]. These include telecommunications infrastructure and technical support, which outperform service recipients through knowledge sharing and health and service quality. An important aspect is creating practical solutions when resources are limited. A functioning support system strongly integrates local communities. It improves the ability to solve issues at the local community level with the substantive support of professionals—doctors, medical professionals, IT specialists, etc. [54,57].

Creating a virtual support system is a viable and sometimes the only alternative for the elderly. The involvement of volunteers, including family members of those in need of care, goes hand in hand with shaping the entire medical support system that uses volunteers at the micro level.

Among the most frequently demonstrated problems is the need for more utilisation of the experience and knowledge of patients and suppliers, and it concerns the planning and design phase. As a result, there needs to be a better match at the necessary competencies and capabilities level. Practice shows that complications, inadequate support or lack of appropriate care are noted for more complex cases [50,52,53]. All responses resulting from the research can be found in Table 4.
ijerph-20-02416-t004_Table 4Table 4Cocreation matrix for EMSS.Actors/PhasesProvidersPayersSuppliersConsumersPlanningProposes new services based on integrated resources.Budgetingcontrolling

DesignHealthcare service provides the design of the system. Mobile company and technical support agent to design the system


DeliveryProvides service quality and accessibilityprovides eHealth, an active service systemMonitor resultsFamily members’ health monitoringData from patients delivered utilizing mobile technology is necessary for health monitoringthe report health status for providers using suppliers’ supportMaintenanceEnable the exchange of datatechnical support systems
Enable to exchange of dataReports suggestions to improveSource: own elaboration.


#### 4.1.2. Case #2: Digital Health Platform Paginemediche.it

##### Description

This example was used to present an analytical tool such as the cocreation matrix in our team’s article entitled “Co-creation in Health 4.0 as a new solution for a new era”. That article was a theoretical analysis of cocreation in Health 4.0, including a literature review. In contrast, this article presents the practical use of the cocreation matrix and verifies the theoretical view of cocreation.

Paginemediche.it is one of the first digital health platforms designed and established in Italy. Such platforms support the flow of knowledge and information between healthcare providers, patients, doctors and caregivers [58]. Paginemediche.it is a platform that offers personalised health services to patients who, either through direct contact via the platform or through the consultation and information retrieval option available on the platform, interact with doctors, participate in therapy or acquire the knowledge and information necessary for treatment or preventive behaviour. In 2022, the number of users registered on the platform exceeded 15 million, while the number of professionals registered in the MediciOnline professional zone and with active accounts exceeded 140,000. Paginemediche.it reports that more than 1 million users per month use the platform’s website for consultations or medical support. It is estimated that Paginemediche.it has become the most important digital tool providing medical services to users in Italy. Moreover, the platform has become the primary source of information on prevention and treatment.

##### Actors

The users of the platform are patients and doctors. The patient activity focuses on building the diagnostic base through search analysis and patient preferences. This builds up the patient’s information in his or her own “Personal Area”, where the patient leaves his or her preferences, information about search queries and goals, and biometric parameters. “Personal Area” is a database about the patient created based on a so-called “digital footprint”; this data is used for analytical and diagnostic purposes. Expanding the database on a mass basis allows the grouping and improvement of diagnostics. Individual data allows better diagnosis and the creation of tailored and personalised health programmes and helps to create a path of medical support, e.g., by arranging the order of visits to doctors or the necessary examinations. In addition, to broaden diagnosis and make it more open, services such as: “Expert Answers”, “Patient Kit”, and “Digital Therapy”. All of these improve the quality of the service, making it more accessible and trustworthy. An additional attribute of the Platform is that it keeps medical history and therapies in one place so patients and doctors can check their medical or treatment history at any time, and patients can check doctors’ recommendations and treatment regimens.

Doctors and medical personnel use the “Professional Zone”. The zone is used to work with patients, obtain information on specific diagnoses, diseases, ailments and therapies, and allow consultation with other doctors and professionals.

The platform makes bilateral use of the activity of patients and doctors. Based on tracking their work and activity, changes can be made to facilitate the use of the platform. The knowledge and experience of medical professionals and physicians are used in all phases of the process. Physicians are heavily involved in building the diagnostic and knowledge base for applied therapies and recommendations. In the case of patients, involvement is decidedly one-sided. Data acquisition is the critical information coming from the patient and is used to profile the platform’s activities. Patients are an essential source of information, and the collection of this information and its use is multi-level. However, the actual involvement of patients and their influence on how the platform operates is negligible.

Patients’ and doctors’ interests are analysed extensively to develop further the platform itself and how doctors and medical professionals work. Interviews point to the introduction of algorithms in diagnostics to help doctors make accurate decisions in this area. There are also opinions on the widespread use of AI for in-depth medical data analysis.

##### Mechanisms/Areas of Cocreation

Cocreation occurs as part of service delivery when patients consult and interact with doctors. At any time, patients can share files online, such as documents and photos, which is entirely secure and can better help the specialist identify their health problems. This creates a space for value cocreation by sharing healthcare experience, information and knowledge.

The coronavirus-derived COVID-19 emergence underscores, much more than before, the need to improve health monitoring with online support, as the Italian blockade and the collapse of the national health system (under pressure from the spread of the pandemic) are limiting citizens’ access to health services. In line with the mentioned arguments, the Paginemediche.it website provides a thematic chat room allowing healthy people to learn how to protect themselves from infection on the one hand and infected patients to be monitored at all times on the other hand. Lo Presti et al. [42] note that such platforms strengthen stakeholder engagement by empowering their users: long-term engagement and relationships are created among users—doctors and patients

An analysis of the table allows noting the low level of payer involvement. The payer outside the planning process becomes only an oversight entity. As a result, it limits the possibility of system development at subsequent levels and the introduction of significant changes due to experience at successive stages of the service development process. As a result, the organisation’s flexibility is reduced, and the ability to effectively use doctors’ and patients’ experiences and opinions is limited. All responses resulting from the research can be found in Table 5.
ijerph-20-02416-t005_Table 5Table 5Cocreation matrix for digital health platform.Actors/PhasesProvidersPayersSuppliersConsumersPlanningCreating solutions based on feedback from doctors and patientsformulating concepts for changing the content of the way of service providedBudgetingcontrollingInformation support, real participation and involvement in the planning processInformation support, but in the form of a passive and one-sided flow of informationDesignHealthcare service provides the design of the system.
Define contents and medical topics as a result of the interaction with doctors—Video Visit Service
DeliveryAnalyse dataConsult patients’ medical data.Enrich own expertise
Provide contents and medical topics to be uploaded into the Platform (medical papers and scientific research)Provide data into the systemsharing the data with members from their networkMaintenanceCreating new solutions and digitally supported services
Building a knowledge asset complementing the content available online and taking care of the quality of services offered.Medical data analysis substantive support of the process of changes and updates in services providedProviding information using direct services and content and services available onlineSource: own elaboration.


#### 4.1.3. Case #3: Electronic Medical Record in Italian Academic Integrated Hospital

##### Description

The Electronic Medical Record is “electronic medical data and reports about patients’ conditions, images, physiological signals, check-up reports, medical treatment videos, and medical forms” [59]. This definition was used by Bonomi et al. [51] to explain the objective of its implementation at AOUI (*Azienda Ospedaliera Universitaria Integrata*) in Verona [60]. AOUI is one of the Veneto region’s largest healthcare providers. Since 2008, AOUI has been undergoing the reorganisation process aiming at the progressive adoption of information systems. Adopting electronic health records (EHR) [61] was the first step of that process. One of the main components of the EHR is the EMR, the repository for all the internal information generated by the respective hospitals’ organisational units.

The system allows people to track and consult their entire health history, sharing it with professionals for more efficient and effective health care. Rapid, centrally supported development means that currently, only three regions have EHR implementation levels below 90% and only one below 50% [62]. An informal study on the implemented system’s effectiveness in two hospitals showed that EHRs improved patient management. Moreover, the software supporting HDL preparation, that is, the document summarising the hospitalisation and including indications for continuing treatment, was found to be the most appreciated functionality [63]. Other positive effects that translated into increasing engagement with the system were an increase in the usability of the base, with studies noting, among other things, increased prescription appropriateness, a reduction in overall nurse workload, and optimisation of medication inventory management. There was also no increase in errors, which is often noticeable in the initial phase of implementation.

Digitalisation allowed for the reduction or replace handwriting, fast identification of patients’ conditions due to access to electronic records and the urge to submit necessary information for subsequent therapy. As Bonomi et al. [51] show, data sharing among different staff members imposes cooperation among them. As a result, we can define the EMR as a repository of patients’ data in a digital form. That system allows hospital workers to unify any patient’s data.

##### Actors

Bonomi et al. [51] note that the value is cocreated by three key actors: physicians, hospital workers and patients. Another paper about the AIH’s digitalisation of radiology was assessed as valuable support of decision-making by reducing the number of errors in tests, improving organisational efficiency, better diagnosis and accurate medical reports [64]. The cocreation study confirmed the growing benefits of increased information sharing among hospital staff as a direct result of cocreation. Notwithstanding the noticeable lack of patients in this list, exchanging medical records was helpful for patients requiring treatment in several hospital units. Research shows that the main driver of system development is pressure from medical professionals, whose involvement in shaping the entire system is the greatest [65]. This is a determining factor in selecting each system implementation’s management path and process.

##### Cocreation Mechanisms/Areas

EHR development began with very close interaction between developers and physicians, adopting the main criteria of an agile development methodology that allows rapid prototyping and continuous user testing of the system. The EMR cocreates value through an endless chain of interactions between service providers and consumers, but in this case, the actual consumers are the hospital staff members. In this case, cocreation consists mainly of knowledge exchange and repository development. The cocreation mechanism is, therefore, visible in the delivery and maintenance phase. Cocreating the system takes place in the form of cooperation between end users, and their knowledge input into the system allows them to improve the quality of work of healthcare workers [65]. In effect of the cocreation, a significant improvement in the decision process may be achieved [51]. During the COVID-19 Pandemic, the EHR was also dedicated to serving and treating SARS-Cov2 patients. This underscores its versatility and flexibility for use in different departments.

The research shows that adopting this product in Italian cities and hospital requires the development of an information society and enhancing the quality of delivery of certain services [66]. This weakness of Italy emerged at both micro and macro level [67].

On the other hand, the lack of support from payers and hospitals in tailoring the system to the needs of hospital operations has meant that changes that expand the clinical and research service department have yet to be made. As a result, one of the essential and expected functionalities has been lost, as it is apparent that EHR effectiveness is not correlated with clinical aspects. Meanwhile, at the same time, apparent differences in mortality rates in emergency and day hospitals are evident between patients in hospitals with “full EHR” or “partial EHR” compared to hospitals with “no EHR” [67].

Other problems include highly bureaucratic implementing institutions as well as the environment. Formalism has become a force that inhibits possible changes. As a result, the introduction of new solutions, even though they save staff time, increase the usability of system operation and improve patient safety, is blocked [65,66]. Another aspect strongly related to the creative model is the limitation of suppliers’ availability in creating a solution. In this case, the role of institutional units is omitted from the process, and the entire process is centrally coordinated by L’Agenzia per l’Italia Digitale and the Ministry of Health [62,65]. The centralisation, in this case, involves combining the role of the regulatory body and the contractor. Medical units are effectively treated as part of the system, and their voices are not heard throughout the process. Interestingly, the opinions of doctors, nurses and other professionals using EMRs are an essential source of information. All responses resulting from the research can be found in Table 6.
ijerph-20-02416-t006_Table 6Table 6Cocreation matrix for EHR in AIH.Actors/PhasesProvidersPayersSuppliersConsumersPlanning
Financial and conceptual support from the government

DesignExpressing a perceived lack of software/ hardware supportFinancial and conceptual support from the government
Improving efficiency and introducing innovationDeliveryCocreation during treatment among various hospital unitssharing information about patients

Complex medical treatmentdata sharing among other consumers (medical staff)MaintenanceConstant compilation of patients’ data, trial and diagnosis

Strengthen trust in hospital-patient relationshipsthe legitimacy of health organizationsSource: own elaboration.


#### 4.1.4. Case #4: TeleAngel (“Małopolska Tele-Angel”)

##### Description

The objective of this project is to improve the quality of life of dependent people through the implementation of activities for the development of care in the place of residence and services using communication technologies (“life band”) that enable seniors to remain safe in their environment [52]. The “life band” monitors the vital functions of the patient and, as such, addresses the field of Health 4.0 and, thus, mobile devices in the field of health protection [68,69]. It provides for remote 24 h medical telecare to be rendered for seniors. The product is a patient-worn device and a telecare service in which an on-call operator, 24 h a day, monitors readings from the wristband. Additional functionality is the ability to use the “SOS button” and connect to a medical centre offering telecare; this way, it is possible to receive support and assistance from professional medical professionals and trained volunteers. It is estimated that more than 6300 used the band continuously in the project’s first edition, and more than 250 people survived in a life-threatening situation thanks to the band’s use.

##### Actors

The project is implemented in partnership created by three types of organisations. The first one is a regional government—the regulator and payer. The next one is Caritas, an institution that coordinates charity activities in this city—the provider. The third partner is a local non-governmental organisation: the European Institute of Regional Development Association—the provider. An IT company—the supplier of medical bracelets called “life band” and the telemedical system is the fourth partner organisation.

##### Co-Creation Mechanisms/Areas

An IT company focuses on developing and delivering a comprehensive service. It also provides technical assistance for the telecare centre, employing medical rescuers. The NGO supports and provides care for the elderly. The NGO knows the client’s expectations. Due to its reputation and credibility, it is more predestined to recruit participants for the project’s purposes and publicise the service among potential beneficiaries. The regional government focuses on project management. Patients provide feedback on the device, its functionality, potential problems, and information about the entire system’s operation. Therefore, the synergy effect is achieved. This form of cooperation allows us to support needy people while testing the new model for providing healthcare services [52] essential for an ageing population. All responses resulting from the research can be found in Table 7.
ijerph-20-02416-t007_Table 7Table 7Cocreation matrix for **TeleAngel**.**Actors/Phases****Providers****Payers****Suppliers****Consumers**Planning
Finance the system

Provide funding policyDesignProvide general guidelines for operating the system and its functionalitiesdesign the systemSets goalsPresent recommendations for service usability
DeliveryProvide servicedata analysisconsultingMonitor resultsRecruiting patients provides supportProvide dataMaintenanceRechnical support systemenable data exchangePromotionSupport program promotional activities reports suggestions for improvementReport suggestions Provide feedbackSource: own elaboration.


## 5. Results and Discussion

The construction of the Matrix makes it possible to demonstrate the involvement of a particular group of actors in a specific phase. As a result, it becomes possible to determine the scale of cocreation for each phase of product or service creation. Figure 1 presents the synthesis of cocreation occurrence within the framework of the four analysed cases. The matrix indicates where cocreation in Health 4.0 appears most often. These areas include the design, delivery and maintenance phases in the case of supplier involvement and the delivery and maintenance phases in the case of consumer involvement, as cocreation appears in all analysed cases.

**Figure 1 ijerph-20-02416-f001:**
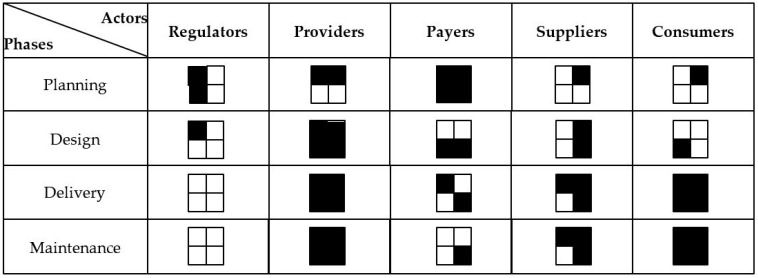
Cocreation in studied cases. Key: the number of black squares represents the number of appearances in all the cases. The left upper square symbolises case #1, right upper-case #2, etc. ⬛—appearance in a given case. Source: own elaboration.

Cocreation in Health 4.0 occurs more often in the delivery and maintenance phases (four cases). A significantly lower number of identified co-creative activities are in design and planning. Health 4.0 product launches involve providers, payers, suppliers and consumers, but only slightly regulators. Few studies confirm this structure of involvement [70].

When analysing individual phases, the most significant actors become apparent in the delivery and maintenance phases. These phases utilise both the demand and supply side of the product and service development process. These phases are characterised by the strong involvement of providers and product recipients, i.e., suppliers and consumers. From the perspective of the creation process’s efficiency, this shows producers’ attempt to use cocreation to mainly affect the distribution and maintenance of the product in the market. In contrast, we see far less use of the potential of cocreation in the product/service creation and planning phases. In these phases, the use of knowledge and experience of doctors, medical staff and patients are marginal, and even the greater involvement of suppliers in the design phase in the cases studied remains the same.

Providers are the key group in the analysed cases and are the initiators of all activities and participants involved in each of the phases of product development and launch.

The last participation of suppliers is evident in the planning phase, whereas payers become the critical actor in this phase. As a result, actors who are not responsible for creation but who are responsible for the financial aspect of the entities’ activities become responsible for creating the framework for the designed solutions. Even if the payer is part of the enterprise and not an independent stakeholder, its voice becomes the most important in the planning phase. At the same time, the role of actors responsible for creation or needs analysis is much more limited. This aspect can be crucial from the perspective of further shaping the entire process of creating a product or service. The heavy involvement of payers in the planning phase limits the flexibility of creation and makes the goals more rigid. As a result, space for extensive collaboration and product creation within the complex cocreation process is impossible, as assumptions and constraints block opportunities for change at later stages, which is the nature of cocreation.

As shown in the literature, suppliers collect their and patients’ opinions during the design process. Based on this, they can present the real expectations and needs of the recipients, in effect guiding the quality and functionality of the created solutions [65]. Those actors in service delivery through specific technology allow for information sharing among the healthcare system’s actors. The next group appearing in the cases in the particular phase is represented by consumers (patients) engaged in service delivery. Their primary role is to deliver data about health conditions and learn about health status. There is significant potential to be exploited by these actors. Entrepreneurs should increase the involvement of these actors and use their knowledge and experience to plan and create solutions in the area of Health 4.0 and beyond. In this way, it is possible to increase the efficiency of the implemented activities and the level of adaptation and, consequently, the use of the implemented solutions.

Suppliers are responsible for providing technical support and recommending how to improve healthcare services. Their primary role in the delivery phase may be to ensure that the system operates and guarantees service availability. A critical factor for both phases is the suppliers’ understanding of customer expectations and needs [71]. In the maintenance phase, suppliers exchange data about services and recommend improvements.

On the other hand, regulators are the least involved actor; they play an essential role in the planning phase for projects and initiatives implemented through public processes (e.g., health or social policy). Regulators play a key role in establishing service delivery frameworks with payers and providers. Regulators are crucial in creating a framework for service delivery with payers and providers.

Applying R. Palumbo’s criteria of breadth and depth of coproduction [1,2], we found out that none of the cases studied represents “group co-creation.” This means that the simultaneous occurrence of high depth and breadth of cocreation does not occur in Health 4.0 services, according to the cases studied. All other types of cocreation occurred. Collective cocreation was identified in case#4 of TeleAngel, while the traditional model was confirmed in case#3 of EMR. Individual cocreation, on the other hand, occurred in case#1 and case#2 (Table 8).

These examples demonstrate the ability to foster individual and group relationships that transform existing healthcare and health service ecosystems. The involved parties can change their roles during the process, and their involvement can influence the actions of other actors in the health system, such as the state, regional and local governments, and private companies [72]. Slight changes in the relationship and the way medical professionals and patients are involved can change the role of patients in the healthcare system: from passive participant to active person who has responsibilities and input into the treatment process, and furthermore, who, at the same time, actively influences the process of creating products and services [1,39,40,73].

## 6. Conclusions

Our work contributes to the existing theory of patient cocreation. In response to the research questions, the scientific literature does not describe many cases of cocreation being used in Health 4.0. The answer to the first research question, “Is cocreation being used in the creation of Health 4.0 products and services?” indicates the limited involvement of patients and professionals (doctors and medical professionals) in the entire process of creating and managing services. However, it is noteworthy that while the issue is not extensively analysed in scientific articles, more extensive information on cocreation can be found in non-scientific articles and case studies of service/product creation. So far, this issue has yet to be widely explored by science. Our analysis shows that even in cases where cocreation has been identified, despite claims by R. Palumbo [1,2] about the potential growing involvement of patient cocreation in the creation process, the identified cases of this do not support this view.

The answer to the second research question, “What is the scope and scale of co-creation in the studied cases of Health 4.0 products and services?” shows that the scale and extent of participants’ involvement in cocreation varies. Providers are most heavily involved in all phases of the cocreation process, while the end users (doctors and patients) involvement is seen mainly in the last two phases. However, by treating this group separately, i.e., singling out suppliers and consumers, a qualitative difference becomes apparent. Both theory and practice reveal greater medical personnel involvement (medical personnel, in some cases, also appear as consumers). Medical personnel are involved more strongly in each phase, while the involvement of patients is seen in the delivery and maintenance phases.

The answer to the third research question, “Do the examined cases differ in terms of how cocreation is used in the process of creating the service/product?” reveals that co-creation is not homogeneous. The way it is implemented may differ due to the product or service and the implementing entity or stakeholders involved. In the analysed cases, co-creation appears as a very flexible way of creating solutions and building a knowledge and experience asset that enriches creating a product or service. In the cases studied, it is more evident that medical personnel are more involved in the whole process. The professional–user relationship is limited to user cocreation of services, as professionals are overwhelmingly service planners [74]. Therefore, the full range of cocreation/coproduction modes and patient empowerment must be visible in Health 4.0 examples.

On the other hand, patients are involved in gaining knowledge in the form of feedback on quality and performance. The analysis results show that consumer involvement in the whole process occurs, but the essential element is the quantitative and qualitative aspects. Patient participation can be assessed as reactive and one-dimensional in the cases analysed. Our analysis shows that even when cocreation occurs in a given case, it can be quantitative rather than qualitative. This means that the use of the process only allows the potential of the whole process to be realised.

It can be concluded that the patient’s existence as a cocreator may occur when: (1) the system has been designed purposefully to engage the customers, (2) customers are engaged in service delivery and concurrently benefit from it, and (3) customers feel empowered and responsible for the results. Those assumptions explain why patients do not appear in the first stages of service management phases but mainly in the last one. Our work could be helpful for Health 4.0 designers for planning projects and stakeholders’ inclusion. The appearance of actors and related intensification volume in Health 4.0 service phases have been demonstrated to vary. It recommends planning in which service management phase each stakeholder should be included and how to use stakeholders’ competencies and experience.

Analysis of the small number of cases found in the indexed scientific literature was a significant limitation of this study. The lack of description in all fields of the cocreation matrix may be due not only to non-existent relationships but to an information gap. The limited number of cases classified as cocreation in Health 4.0 calls for investigating new cases to contribute to and enrich meta-analytic studies.

## Figures and Tables

**Table 1 ijerph-20-02416-t001:** The service life cycle’s development.

Phase	Description
Planning	The essential elements of this phase include initiation, feasibility study, risk, strategy, schedule and budget [35,46].
Design	This phase addresses the problems associated with creating a product, system and technology [39,46].
Delivery/implementation	That is the most challenging phase; many companies need help with completing this phase [46]. The critical element of this phase is assessing test results and planning the procedure [47].
Maintenance	During this phase, suppliers must support the operational effectiveness of the product/service. It requires constant monitoring, corrections, updates and troubleshooting [46].

Source: own elaboration.

**Table 2 ijerph-20-02416-t002:** Classification of healthcare actors.

Actor	Description
Regulators	The ministry of health and those in charge of health issues at the regional and local levels are regulators.
Providers	Providers can be treated as supply and demand sides simultaneously. On the other hand, they offer solutions and consume the data and products [35,49]. That group comprises doctors, nurses and entities such as hospitals or care homes [40].
Payers	The payers create statutory and private health insurance. However, their role is significant for maintaining the system’s operations due to the financial system. The strongly differentiated part of payers depending on the healthcare system is essential in this case.
Suppliers	The suppliers facilitate the service provision and embrace pharmacies, pharmaceutical companies, medical device companies and ICT companies. Suppliers are often the initiators of the whole process and creators, beginning with the planning phase until successful implementation and achievement of deliverables.
Patients	The largest group in this collation, although they have the weakest influence on the system. The patient is the end user in the product creation process. From a market perspective, the patient creates the demand for Health 4.0 products.

Source: own elaboration.

**Table 8 ijerph-20-02416-t008:** Depth and breadth of cocreation.

Case ID	Case Name	Depth of Cocreation	Breadth of Cocreation	Type
Case #1	Paginemediche	High	Narrow	Individual
Case #2	EMMS	High	Narrow	Individual
Case #3	EMR	Low	Narrow	Traditional
Case #4	TeleAngel	High	Broad	Collective

Source: own elaboration.

## Data Availability

Not applicable.

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
