# Peer review of "The Stakeholders’ Involvement in Healthcare 4.0 Services Provision: The Perspective of Co-Creation"

_ijerph, 2023, doi:10.3390/ijerph20032416_

Round 1
Reviewer 1 Report
Manuscript “The scope of Stockholders’ involvement in Health. 4.0 adoption: The Perspective of co-creation” is critically reviewed and found an interesting topic towards health sector but needed following amendmentsTopic:
I suggest a revision in title of the manuscript to make it representative. Abstract:
The opening sentence makes no sense of understanding followed by a baseline which is not written good enough to support the study; no proper brief methodology defined. No results. Abstract is needed to be rewritten. Introduction:
There are lots of paragraph in this section but no alignment within paragraphs/information was established. Introduction should be brief, concise with updated supporting material.
Methods
As our initial analysis… Where it is defined? Unfortunately, methodology is not sufficient and relevant.
References:
Check your references carefully and formatting according to the journal outline. Overall manuscript needs to improve by removing grammatical mistakes.
Reviewer 2 Report
I recommend that this paper be accepted after revision.
1. In the “Introduction” section, the last paragraph is unnecessary, and it is recommended that it be deleted or simplified in the article.
2. In the section 2, since it is mainly a review of the literature, "Co-Creation in Health 4.0" would be more appropriately replaced by "Literature Review".
3. In the section 3, "Method" should be changed to "Methodology" because “Method” emphasizes the behavior or tools used to select a research technique, while “Methodology” is analysis of all the methods and procedures of the investigation. In addition, it is also necessary to add the detailed reasons for the selection of cases in this section.
4. The discussion and conclusion sections are not sufficiently well presented. In the "Discussion" section, it is suggested that a comparative analysis of the four cases be combined to make the in-depth discussion more convincing. Please address these issues.
Reviewer 3 Report
The theme of this study is highly relevant in todays context and is within the scope of IJERPH. Major revision is suggested to further enhance the content quality.
Rewrite the Introduction section. Clarify the need of this study?
What are the objectives of this study?
Strengthen the literature review section. Refer recent and relevant papers.
Explain the research methodology properly.
Rewrite the case presentations in professional manner.
What is the contribution of this study?
Rewrite the Conclusion section. Focus on novelty, limitations and future scope
Reviewer 4 Report
The authors determine the scope and depth of the co-creation of health 23 services based on modern technological solutions (Health 4.0). they selected 4 cases involving Health 24 4.0 solutions, verified the scale and scope of co-creation using them as examples, and used the Co- 25 creation Matrix to do so. There are a few comments that should be considered to improve the paper
1-in line 87, health services (see: [removed for review]) using technological solutions related to the, what do you mean by (see: [removed for review])
2-contribution should be highlighted in the introduction section
3-Authors discussed Industry 4.0 and healthcare 4.0, but many recent works are not discussed and taken into consideration such as Autonomous Multi-Robot Collaboration in Virtual Environments to Perform Tasks in Industry 4.0, Industry 4.0 towards Forestry 4.0: Fire detection use case, Digital twins collaboration for automatic erratic operational data detection in industry 4.0, Computing in the sky: A survey on intelligent ubiquitous computing for uav-assisted 6g networks and industry 4.0/5.0,Personal digital twin: a close look into the present and a step towards the future of personalised healthcare industry,Shape memory alloy-based wearables: a review, and conceptual frameworks on HCI and HRI in Industry 4.0, and etc.
3-explain Digital Health Platform. with more details
4-explain table 5 with more details
5- authors added references in conclcusion, i suggest authors focus in summaries their contribution and if they need to do compersion, it should be moved to discussion section
Round 2
Reviewer 3 Report
The theme of this study is relevant in todays context. However, major revision is suggested to further improve it.
Literature review section needs to be strengthened. Many papers have been published in healthcare sector. Author/s may refer relevant papers.
Rewrite the cases in professional manner.
Justify the results? Rigourous analysis is expected. Use logical approach
Whether results are drawn from qualitative analysis or quantitative data? Clarify?
Highlight the novelty of this study
Rewrite the Conclusion section accordingly
Author Response
Please see the attachment.
|
Reviewer 3 |
|
|
Rewrite the Introduction section. Clarify the need of this study? What are the objectives of this study? |
We have revised and rewritten the introduction. The purpose of our article and the research questions were clearly outlined in the first chapter. |
|
Strengthen the literature review section. Refer recent and relevant papers. |
The literature review has been expanded and revised. It is worth emphasising that this article is a continuation of the topic undertaken in the article "Co-creation in Health 4.0 as a new solution for a new era," in which our team carried out a theoretical analysis of co-creation in Health 4.0 based on a broad review of the literature. In that article, a co-creation matrix was proposed as an analytical tool to assess the scale and scope of co-creation. The reviewed article attempts to verify this tool in practice and answer how the scale and scope of co-creation in Health 4.0 are shaped. |
|
Explain the research methodology properly. |
We followed the recommendation of SLR and explained undertaken steps. |
|
Rewrite the case presentations in professional manner. |
The case presentation was restructured, improving the description of the main sections derived from the Co-Creation Matrix.. |
|
What is the contribution of this study? |
The purpose of our article and the research questions were clearly outlined in the first chapter, and the contribution of our research was more clearly outlined in the last chapter. |
|
Rewrite the Conclusion section. Focus on novelty, limitations and future scope |
The conclusion chapter has been revised. The chapter includes a summary of the entire issue, limitations, and areas for future analysis. |
